# Cascaded 2D Micromirror with Application to LiDAR

**DOI:** 10.3390/mi14101954

**Published:** 2023-10-19

**Authors:** Behrad Ghazinouri, Siyuan He

**Affiliations:** Mechatronics and MEMS Research Laboratory, Toronto Metropolitan University, Toronto, ON M5B 2K3, Canada; bghazinouri@torontomu.ca

**Keywords:** 2D micromirror, cascaded, LiDAR

## Abstract

This paper introduced a novel approach to enhance the vertical scanning angle of a large aperture 2D electromagnetic micromirror through the utilization of a cascaded torsional beam design. The primary objective was to increase the vertical scanning angle without compromising the robustness, which was achieved by optimizing the trade-off between the rotation angle and the first mode of resonant frequency. The cascaded design provides flexibility to either increase the outer frame’s rotation angle without sacrificing torsional stiffness or enhance the torsion beam’s stiffness while maintaining the same rotation angle, thus elevating the first-mode resonant frequency and overall robustness. The effectiveness of the cascaded design was demonstrated through a comparative study with a non-cascaded 2D micromirror possessing the same aperture size, torque, and mass moment of inertia. Theoretical analysis and finite-element simulation are employed to determine critical parameters such as the stiffness ratio between the cascaded torsion beams, and to predict improvements in the scanning angle and primary resonant frequency brought by the cascaded design. Prototypes of both cascaded and non-cascaded designs are fabricated using a flexible printed circuit board combined with Computer numerical control (CNC) machining of a Ti-alloy thin film, confirming the superior performance of the cascaded 2D micromirror. The cascaded design achieved vertical scanning angles up to 26% higher than the traditional design when both were actuated at close resonance frequencies. Additionally, the micromirror was successfully integrated into a 3D LiDAR system. The light detection and ranging (LiDAR) system was modelled in Zemax OpticStudio to find the optimized design and assembly positions.

## 1. Introduction

The rapid advancements in autonomous vehicle technology have significantly increased the need for a dependable light detection and ranging (LiDAR) system [1,2,3]. With the increasing need for cost-effective and compact LiDAR solutions, micromirror-based LiDAR technology emerges as a promising candidate to replace its bulkier and pricier motorized counterparts. Micromirrors offer a smaller form factor and cost advantage, addressing two key drawbacks of motorized LiDAR systems. However, certain aspects of micromirror-based LiDAR require enhancement to outperform motorized counterparts fully [1,4].

Micromirrors have been used in applications such as projection display [5,6] augmented reality display [7], and medical imaging [8]. But their integration into a LiDAR system introduces a specific requirement which is different from the previous applications. While in previous applications, the micromirror functioned solely as a transmitter, in a micromirror-based LiDAR, the reflected light from the target should be redirected toward the receiver sensor, and the small aperture size of the micromirrors designed for other industries can limit the amount of light received by the receiver. As a result, this limitation in aperture size can subsequently restrict the maximum detectable distance for a LiDAR system. Various strategies were adopted to address this challenge.

One approach is to incorporate a beam splitter between the micromirror and the transmitter/receiver [9]. However, the presence of a beam splitter in the light path can diminish the power of received light, potentially interrupting the detection of low-power reflections. Another approach is to use an array of receivers facing the target, leaving the micromirror solely responsible for transmitting the laser beam. However, this setup is more expensive and requires a more complicated assembly process and processing algorithm [10]. The third approach is to use the micromirror as both a transmitter and a receiver of light. This method demands a larger aperture size for the micromirror, as the received signal power is directly proportional to the receiver’s optical aperture [4,11].

Compared to alternative methods, a large aperture micromirror offers a simple and cost-effective solution. To address this, the authors’ research group introduced flexible printed circuit board (FPCB) micromirror technology [12,13,14,15]. While FPCB micromirrors provide the benefit of large apertures with a low cost, they suffer from low torsional stiffness (then, low primary resonant frequency), impacting the overall system robustness.

A large aperture electromagnetic 2D gimbal micromirror (Figure 1) was previously developed by the authors for LiDAR applications [16]. The permanent magnets generate Lorentz forces on two distinct current-carrying coils located under the mirror (hereafter referred to as mirror coils) and under the frame (hereafter referred to as frame coils), enabling 2D actuation of the micromirror.

The robustness of such a micromirror against external vibration is determined by the first mode of the resonant frequency of the micromirror [17,18]. The challenge associated with that mirror is that in order to increase the resonant frequency, the torsional stiffness should be increased. Consequently, a larger actuation force is required to achieve the desired scanning angle (and then the field of view (FoV)). As a result, there will be a trade-off between the robustness and the FoV. In order to be able to examine the performance of micromirrors, Wang et al. [4] defined a figure of merit (*FoM*) as Equation (1), which incorporates three critical factors to be considered in the design of a 2D micromirror for LiDAR applications.
(1)FoM=θ×f×d
where θ represents the effective optical FoV, which is the geometric mean of the horizontal and vertical optical angles in radians, f is the effective resonance frequency (kHz), which is the geometric mean of the horizontal and vertical resonance frequency, and d indicates the effective dimension, also known as the aperture size, of the mirror plate in millimeters (mm).

In this paper, the concept of cascaded torsional beams is adapted to the FPCB micromirror frame to help increase the scanning angle or increase the primary resonant frequency (and then the robustness) without lowering the scanning angle. Building upon the previous cross-talk free FPCB 2D micromirror [16], the primary objective of the design in this paper is to increase the vertical FoV without compromising the robustness. The cascaded torsional beam design provides the flexibility to optimize the trade-off between the optical angle, i.e., the FoV, and the first mode of the resonant frequency. This optimization can either increase the outer frame’s rotation angle without sacrificing torsional stiffness or enhance the torsion beam’s stiffness while maintaining the same rotation angle, thus elevating the first-mode resonant frequency of the frame and achieving greater overall robustness.

A theoretical analysis and a simulation are conducted in adapting the cascaded design to the 2D micromirror in order to determine parameters such as the stiffness ratio between the two cascaded torsion beams’ stiffnesses, as well as predicting the improvement in the scanning angle and primary resonant frequency brought by the cascaded design. Prototypes are fabricated to experimentally verify the superiority of the cascaded 2D micromirror.

The paper is organized as follows. Section 2 presents the design and optimization details of the 2D micromirror. Prototyping and testing are given in Section 3, Section 4 and Section 5. Conclusions and discussion are presented in Section 6 and Section 7.

## 2. Design

Cascaded torsional beams have previously been employed in micromirror designs [9,19,20,21], with the primary objective of eliminating the presence of current-carrying wires on a torsional beam, particularly in 1D micromirrors. In the 2D design adaptation of this method, at least two of the torsional beams should contain current-carrying wires. Additionally, this design enables higher torque generation in micromirrors with smaller aperture sizes and prevents the mirror from heating by separating the coiled structure from the aperture. In those 2D micromirrors, the torsional beam stiffness of the outer frame significantly exceeds that of the inner frame, and only the outer frame is coiled to contribute to the actuation force. As a result, this design is limited to resonance actuation.

In contrast to prior applications, the micromirror proposed in this paper employs a cascaded design to increase the actuation angle of an FPCB 2D micromirror without compromising its primary resonance frequency, thus preserving its robustness. In this novel design, the torsional stiffness of the outer torsional beam is not significantly higher than that of the inner beam. Furthermore, both the inner and outer frames are coiled, contributing to the actuation by twisting the two cascaded torsional beams.

In summary, the key distinctions between the suggested cascaded design and the earlier utilization of cascaded designs are as follows:In the prior cascaded designs, only the inner torsional beam is responsible for actuation, with the outer torsional beam remaining significantly stiffer and nearly stationary. In contrast, the proposed design involves both torsional beams in actuation, and the ratio of torsional stiffness between them is considerably less significant and under control.Previous cascaded designs generated actuating torque solely on the outer gimbal, whereas in the proposed design, both gimbals are utilized to generate torque.While the earlier cascaded designs were limited to resonance actuation, the proposed cascaded design offers the flexibility of actuation in either resonance or quasi-static modes.Most importantly, the problem addressed by the proposed design differs from that of previous methods. Previous cascaded designs aimed to reduce the number of torsional beams containing current-carrying wires and increase torque for small micromirrors alongside heat control in the reflective layer. In contrast, the proposed cascaded design focuses on optimizing the trade-off between resonance frequency and actuation angle for large aperture micromirrors.

### 2.1. Theoretical Analysis

In a traditional non-cascaded design, enhancing the generated torque necessitates increasing the coil’s number of turns within the micromirror. Assuming that the coil density is constant, the volume should increase in order to add more coil to the system. This augmentation also increases the mass moment of inertia, causing a decrease in the resonance frequency. As shown in Figure 2a, through the presented cascaded frame approach, a similar technique for boosting torque can be applied. In other words, due to the increased torque, the outer frame rotation stiffness can be increased without lowering the outer frame’s rotation angle, which leads to higher first-order resonant frequency (i.e., better robustness).

To analyse the performance of the cascaded design, a traditional gimbal 2D micromirror (Figure 2b) with equivalent torque and mass moment of inertia was developed (hereafter, it is called traditional design). Thus, the traditional design would have the same first-order resonant frequency as that of the cascaded design proposed in this paper. The objective is to demonstrate that the cascaded design yields a larger outer frame rotation angle than the traditional design. The inner frame’s actuation characteristics of the two micromirrors are identical and are not the primary focus of this study, whose additional details can be found in [16].

In a traditional design, as shown in Figure 2b, the torque generated by the Lorentz force on the frame coils (*T*_1_ + *T*_2_) induces torsion in the outer torsional beam (*k*_3_), resulting in a vertical tilt of the structure to an angle $\theta_{3}$. In the proposed cascaded design, as shown in Figure 2a, the frame coil is divided into two segments, and an additional torsional beam (*k*_2_) is introduced between them. This arrangement causes torques that induce torsion in both torsional beams. The vertical tilt angle of the inner frame ($\theta_{2}$) essentially represents the vertical tilt of the mirror.

The equations of first-order resonant frequency ($f_{1}$ for the cascaded and $f_{3}$ for the traditional design) and the vertical mechanical static half-angle ($\theta_{2}$ for the cascaded design and $\theta_{3}$ for the traditional design) of the two micromirrors are listed in Table 1.

The damping effect was not considered in the analysis of this section to simplify the comparison. This is because both designs share a similar damping ratio, given their identical mirror size, torsional beam material, and comparable torsional beam dimensions. To compare the two designs, the resonant frequency was expressed as a function of the mechanical static angle, eliminating the torsional stiffness(es) as a common variable while considering torques and mass moments of inertia as constants.

While formulating this equation for the traditional design is straightforward, for the cascaded design, it proved to be more complex due to the involvement of two torsional beams, leading to a more complicated resonant frequency equation. As a result, a new variable *x* was proposed as the ratio of stiffnesses of the two cascaded beams (i.e., *x* = *k*_2_/*k*_1_), and MATLAB was used to solve the equation. As a result, the resonance frequency and static angle of the two designs can be defined as two functions, denoted as $f_{1}(\theta_{2},x)$ for the cascaded design and $f_{3}(\theta_{3})$ for the traditional design. By comparing the two designs, it can be concluded that a cascaded design with *k*_1_ = *k*_3_ and *k*_2_ = ∞ is essentially a traditional design. In other words, $f_{3}(\theta_{3})$ is a boundary condition of $f_{1}(\theta_{2},x)$ when *x* → 0. Therefore, if *x* is optimized to achieve the highest static angle for a specific resonance frequency, the cascaded design will undoubtedly achieve a greater angle than the traditional design.

To address the objective mentioned above, the goal of this calculation is to show that if $f_{1}=f_{3}$ then $\theta_{2}>\theta_{3}$ or if $\theta_{2}=\theta_{3}$ then $f_{1}>f_{3}$, which indicates that if the torsional stiffness values for both designs were chosen to maintain identical resonance frequencies, the vertical scanning angle of the cascaded design would be greater than that of the traditional design, or if the vertical scanning angles of both designs are chosen to be identical, the cascaded design would have higher stiffness and then a higher first-order resonant frequency, i.e., better robustness.

In this section, only static vertical scanning angles of the two designs are calculated for comparison without considering the resonant scanning angles under the assumption that both designs of the 2D micromirror employ the same manufacturing process, resulting in similar damping ratios; and larger static vertical scanning angles lead to higher resonant vertical scanning angles for both traditional and cascaded designs.

In order to achieve the maximum mechanical static half-angle for any particular resonance frequency, the frequency *f*_1_ was written as a function of *θ*_2_ and *x*, while the other variables are considered to be constant. Solving ∂*f*_1_/∂*x* = 0 and substituting the local extremum into the function gives out the optimized *f*_1_ as a function of *θ*_2_. Equation (2) shows the optimized value of $x_{opt}$ computed using MATLAB. This equation demonstrates that when the torque and mass moment of inertia values are held constant, the optimized ratio of the torsional stiffnesses remains unchanged. Therefore, if the torsional stiffnesses increase (while maintaining the same ratio), the resonant frequency and mechanical static half-angle will change, but the design will remain optimized.
(2)$$x_{opt}=\frac{T_{1}I_{2}^{2}\sqrt{\left.T_{2}(T_{1}+T_{2}\right)}+T_{2}I_{1}I_{2}\left(T_{1}+T_{2}\right)-T_{2}I_{1}I_{2}\sqrt{\left.T_{2}(T_{1}+T_{2}\right)}}{\left(T_{1}+T_{2})(T_{2}I_{1}^{2}+2T_{2}I_{1}I_{2}-T_{1}I_{2}^{2}\right)}$$

When $x_{opt}$ is substituted into the *f*_1_ equation, it becomes exceedingly complicated and impractical to solve symbolically in MATLAB R2021a. Consequently, it was not feasible to demonstrate that this local extremum represents the global maximum. Therefore, a range of torques and mass moment of inertia were applied to the equation to determine whether $\theta_{2}>\theta_{3}$ for $f_{1}=f_{3}$. This range of values was chosen based on the modelling and simulation values and will be discussed in Section 2.2.

### 2.2. Modelling and Simulation

To determine the numerical torsional stiffness using Equation (2), the mass moment of inertia and torque values were calculated through Solidworks modelling and Ansys Magnetostatic simulation, respectively. The micromirror model includes FPCB with embedded copper coils, Ti-alloy for reinforcement, and a silicon plate as the reflective layer. Further details on these components can be found in the fabrication section of this paper. By utilizing the calculated values provided in Table 2 and applying them to Equation (2), the optimized ratio was calculated as $x_{opt}=0.32$.

By incorporating a range of different torques into Equation (2) through simulation, it was observed that different torques do not affect the $x_{opt}$ and only the ratio of *T*_2_/*T*_1_ changes it significantly. Figure 3 shows the frequency vs. angle diagram for the cascaded and traditional design for a range of torques around the torque values of the simulated model with different *T*_2_/*T*_1_ values. For all these torque values, *T*_2_ was kept constant, and the ratio was changed. In all cases, $\theta_{2}>\theta_{3}$ for any resonance frequency. The same approach was applied to various mass moments of inertia, and the results consistently demonstrated that the $x_{opt}$ calculated from Equation (2) consistently yields a larger scanning angle.

By having the optimized ratio of the torsional beams for the cascaded design, the torsional beam dimensions for both the cascaded and traditional designs can be calculated to achieve any desired resonant frequency. In this study, a resonant frequency of >100 Hz was targeted, which is about 4 times higher than the previously designed micromirror [16]. An initial estimation of the torsional beam dimensions was calculated using rectangle torsional beam equations provided by Young et al. [23], and then, Ansys Modal analysis was employed to determine more accurate torsional beam dimensions for both designs.

The modal analysis results of the cascaded design and traditional design are presented in Figure 4 and Figure 5, respectively. Based on the modal shape results, the first mode of actuation for the frame is 125.83 Hz for the cascaded design and 121.33 Hz for the traditional design. Moreover, in the cascaded design, the mirror actuation occurs in mode 4 at 291.25 Hz, while in the traditional design, it occurs in mode 3 at 298.31 Hz.

## 3. Fabrication

In 2016, the author’s group introduced the FPCB micromirror technology [24] and has since been developing various FPCB micromirrors [14,15,25,26]. While FPCB manufacturing offers advantages such as a multilayer coil, a high fatigue life [27], and a commercialized manufacturing method, the low Young’s modulus of the material poses limitations on the stiffness and then the low first-order resonance frequency of micromirrors, which also represents the low robustness. In 2019, Zuo and He developed a one-step etching method [28] to enhance a 1D micromirror’s robustness by adding a layer of silicon. In this paper, the issue is addressed through the utilization of the cascaded design, a solution that enhances the micromirror’s robustness. Furthermore, a cost-effective proof-of-concept method is introduced, which is particularly well-suited for rapidly validating novel concepts in university research labs. This approach uses traditional Computer numerical control (CNC) machining, eliminating the need for expensive silicon wafers manufacturing techniques in a cleanroom through employing cost-effective materials such as flexible printed circuit boards (FPCB) and thin titanium films, as well as conventional CNC machining technology.

Previous research conducted by Ye et al. [29] demonstrated that the TC4 Ti-alloy is a superior metal compared to SUS304 steel and 7050 Al alloys when used as a stiffener in a 1D large aperture micromirror. The advantage of using Ti-alloy is that manufacturing a Ti-alloy enforced FPCB micromirror does not necessitate complex cleanroom machinery, making it a practical method for producing prototypes for a fast proof of concept.

Using the cascaded design in this scenario is particularly advantageous. This is because the incorporation of the Ti-alloy reinforcement substantially enhances the resonance frequency. However, this enhancement comes at the cost of a reduced optical scanning angle, particularly in the vertical scanning dimension.

A prototype of both traditional and cascaded designs was fabricated and subjected to performance testing. The prototypes comprised five main components: (1) An FPCB base featuring four layers of embedded copper coils. As illustrated in Figure 6a, the coils of the frames were interconnected in series and occupied two layers, while four coil layers were embedded beneath the mirror. This component was fabricated by a commercial PCB manufacturer (PCBWay, KLN, Hong Kong). (2) A gold-coated silicon layer shown in Figure 6b was produced in an ISO class 7 cleanroom (University of Toronto cleanroom). The layer was created using an evaporation technique to deposit a 100-nanometer gold coating on a 200-micrometer-thick silicon wafer, which was later diced into 19 mm × 19 mm squares using a silicon dicing saw. (3) A 0.81 mm thick TC4 Ti-alloy reinforcement is shown in Figure 6b, which was manufactured using end mill CNC machining. (4) Six block magnets made of N52 neodymium. (5) A 3D-printed case structure. These components were assembled to create the prototypes as depicted in Figure 7. The Ti-alloy frame was attached to the FPCB structure using Tesa 8853 tape, which had been applied to the FPCB by the manufacturer. Following this, the silicon layer was bonded on top of the FPCB structure using a cyanoacrylate adhesive.

The resistance of the mirror coil and frame coil of the cascaded design prototype are 81.3 Ω and 117.1 Ω, respectively, while the resistance of the mirror coil and frame coil of the traditional design prototype are 81.5 Ω and 117.3 Ω, respectively.

## 4. Testing

The optical FoV of the two prototypes, actuated at resonance with 5 V amplitude for vertical actuation, was compared and is depicted in Figure 8. The frame resonance frequency (vertical actuation) of the cascaded design prototype and the traditional design prototype were observed to be 134.4 Hz and 103.7 Hz, respectively. The mirror resonance frequency (horizontal actuation) of the cascaded design prototype and the traditional design prototype were observed to be 290 Hz and 294.7 Hz, respectively. Figure 9 shows a comparison between the voltage and angle diagram of the prototypes actuated at resonance.

Based on the data presented in Figure 9, the cascaded design achieved vertical optical angles (e.g., outer frame scanning angle) that were 18% to 26% higher than the traditional design when both were actuated at resonance.

According to Table 1, the relation between the resonance frequency and vertical static angle is $f_{3}\propto \sqrt{\frac{1}{\theta_{3}}}$. This implies that the 23% difference in prototype resonance frequency can result in a substantial 62% reduction in the maximum static angle. Therefore, the improvement provided by the cascaded design will be even more significant in a fairer comparison.

Figure 10 shows a Lissajous pattern created by the cascaded design when it is actuated at double-resonance. With the improved performance of the cascaded design confirmed, it was ready to be utilized in a LiDAR setup.

The temperature variation of the setup was recorded across 7 million actuation cycles at 290 Hz, 5 V using the infrared thermometer GM1350. These data were collected to assess whether the temperature might reach a level that could impact the mirror’s curvature. As depicted in Figure 11, the temperature stabilizes after 1 million cycles at 24 °C ± 0.1, which is very close to the room temperature.

## 5. Application of the Cascaded 2D Micromirror to LIDAR

The micromirror was subsequently integrated into a custom-made LiDAR setup, as shown in Figure 12a. The LiDAR setup was also modelled in Zemax OpticalStudio in order to examine the performance and find the optimized aperture size and position of the LiDAR. More details are provided in the Appendix A.

The Benewake TF03 served as the single-point LiDAR and Teensy 3.5 was used as the microcontroller. The scanning data were demonstrated as a point cloud in Processing 3 software (https://processing.org/). This setup was previously used with two other micromirrors [14,16] that had a larger field of view (FoV) and a vertical actuation frequency of 1–5 Hz. Given that the TF03 frame rate is 10 kHz, the broader FoV in the prior designs led to a reduction in angular resolution. As a result, in order to achieve a point cloud which clearly depicts object boundaries using those micromirrors, the maximum distance had to be limited to ~10 m.

To prevent any additional decrease in angular resolution resulting from the considerably faster actuation of the current mirror, a 5 V actuation voltage was implemented for both vertical and horizontal actuation at resonance. As shown in Figure 12b, the combination of the Lissajous scanning pattern, large aperture size, and smaller FoV, 16.3° ± 0.7 horizontal × 4.4° ± 0.5 vertical, enabled us to achieve more distinct object boundaries in distances of 20 m and beyond.

## 6. Summary and Discussion

To contrast the proposed design with other large aperture micromirrors in the literature, a comparison of their respective FoM values was conducted using Equation (1). Based on the data presented in Table 3, the FoM of the cascaded design demonstrates a notable enhancement when contrasted with the traditional design. Furthermore, when compared to the previous design [16], the FoM displays a substantial improvement of 62%.

While it can be argued that the gap in the traditional design frame can be filled with extra coils, it is important to note that due to the proportion of open space to the entire frame area, the extra coils introduced in that section would only contribute to less than 10% of the torque. However, this approach would also lead to an increase in the mass moment of inertia, which negatively impacts the resonant frequency. Therefore, in the traditional design, the gap was deliberately added to ensure that both the mass moment of inertia and torque values align with those of the cascaded design.

Furthermore, one could also raise the question of what the outcome might be if the frame were enhanced with three, four, or even more cascaded designs. There are noteworthy concerns associated with such an extension. Firstly, the incorporation of more torsional beams necessitates increased space, which could potentially lead to the coil being positioned outside the most optimal zone of magnetic flux density. Secondly, a significant issue arises with the optimization process. Introducing *n* cascaded torsional beams results in *n* − 1 ratios that require careful optimization, leading to a considerably complex calculation process.

Although the impact of dynamic and static deformation on this design was not investigated in this study, further information regarding deformation and flatness in FPCB-based micromirrors with a 24 mm × 24 mm aperture size can be found in prior research conducted by our research group [28].

Lastly, the FoV for this design was set based on the TF-03 frame rate to extend the detectable range. In order to improve the FoV, it is possible to use a thinner Ti-alloy layer with hollow sections under the mirror, which would lower the moment of inertia. This change would make the torsional beams less stiff and increase the optical angle without affecting the resonance frequency.

## 7. Conclusions

This paper presented the utilization of a novel cascaded design to enhance the FoV of a large aperture 2D electromagnetic gimbal micromirror. The effectiveness of this enhancement was confirmed through analytical validation and experimental confirmation, involving a comparative study with a traditional 2D micromirror possessing the same aperture size, torque, and mass moment of inertia. Furthermore, the integration of the micromirror into a 3D LiDAR system was achieved by coupling it with a commercial single point LiDAR. The Zemax OpticalStudio was employed to model the setup, allowing for the optimization of the micromirror’s position, aperture size, and horizontal FoV to attain optimal performance. While the focus of this work was to improve the vertical scanning angle for a fixed resonance frequency, it is possible to use the same approach to increase the resonance frequency for a fixed scanning angle.

## Figures and Tables

**Figure 1 micromachines-14-01954-f001:**
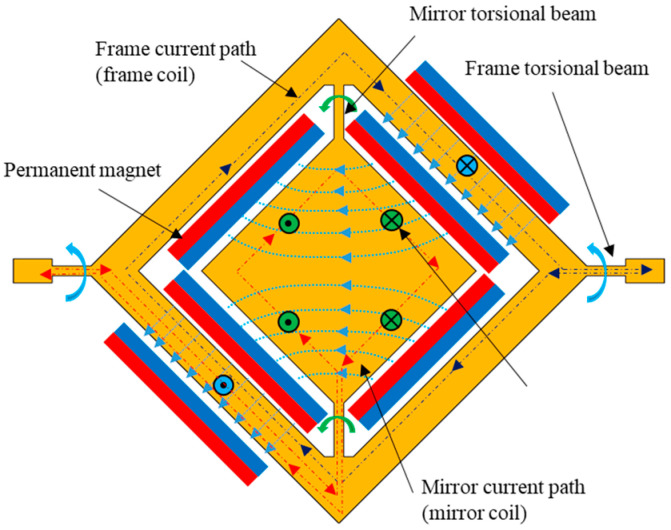
An electromagnetic 2D gimbal micromirror (image from [16]).

**Figure 2 micromachines-14-01954-f002:**
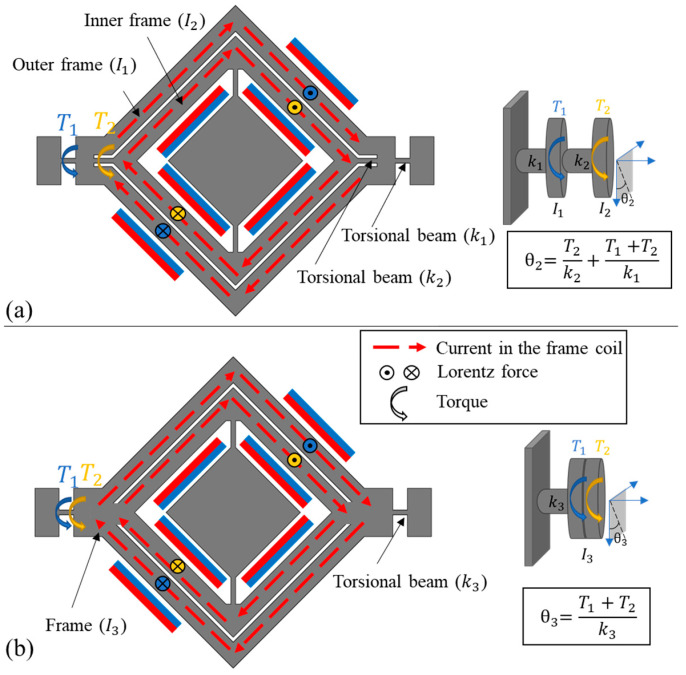
(**a**) Cascaded design and (**b**) traditional design of electromagnetic 2D gimbal micromirror.

**Figure 3 micromachines-14-01954-f003:**
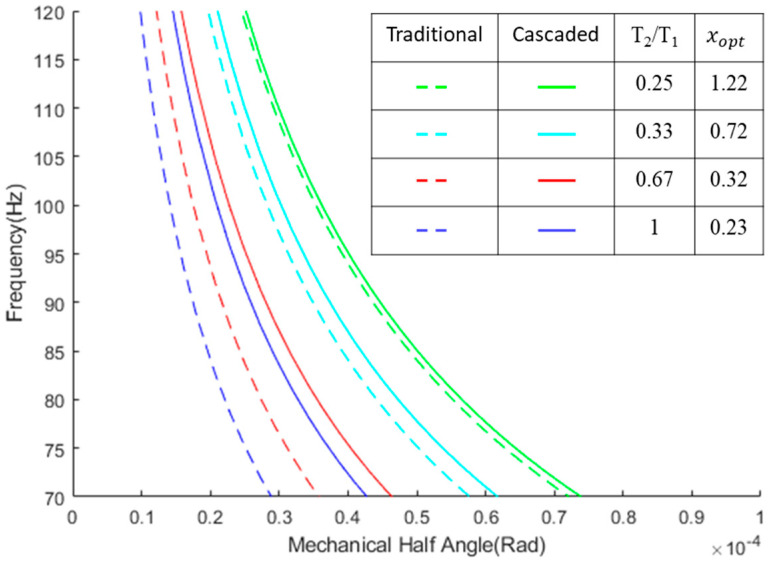
Frequency vs. mechanical half-angle diagram comparing the traditional and cascaded designs across a range of different torque ratios.

**Figure 4 micromachines-14-01954-f004:**
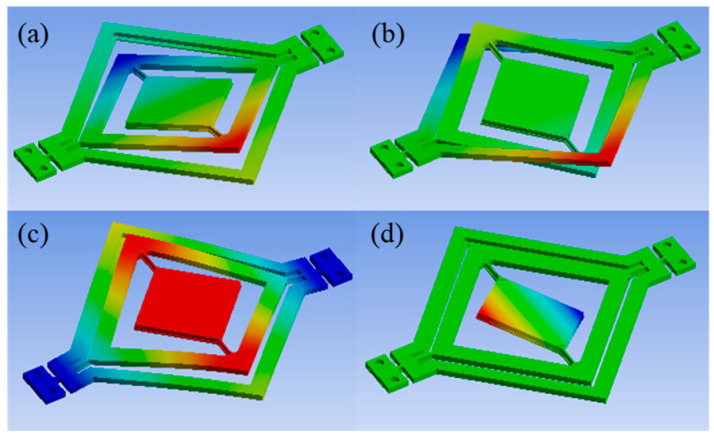
Modal shape results for the cascaded design. (**a**) Mode 1 (vertical actuation): 125.83 Hz; (**b**): mode 2: 244.42 Hz; (**c**): mode 3: 255.46 Hz; (**d**): mode 4 (horizontal actuation): 291.25 Hz.

**Figure 5 micromachines-14-01954-f005:**
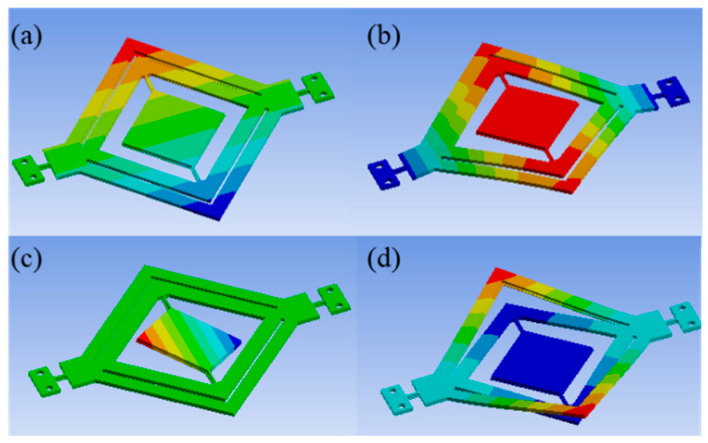
Modal shape results for the traditional design. (**a**): Mode 1 (vertical actuation): 121.33 Hz; (**b**): mode 2: 236.67 Hz; (**c**): mode 3 (horizontal actuation): 298.31 Hz; (**d**): mode 4: 491.51 Hz.

**Figure 6 micromachines-14-01954-f006:**
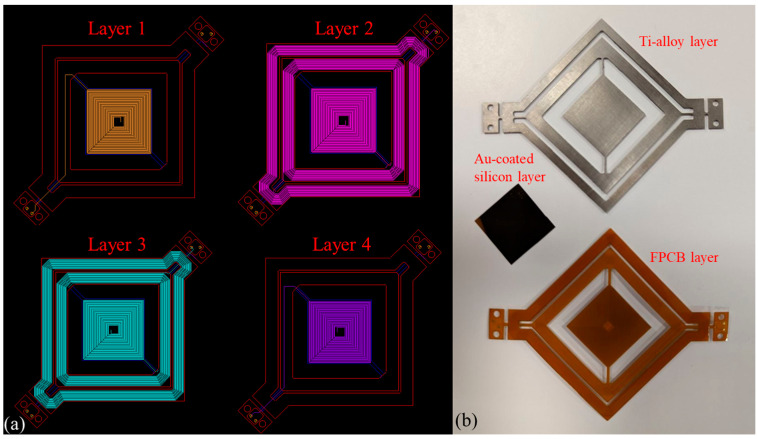
(**a**) Embedded coil layers in the FPCB base. (**b**) Layer components of the cascaded micromirror.

**Figure 7 micromachines-14-01954-f007:**
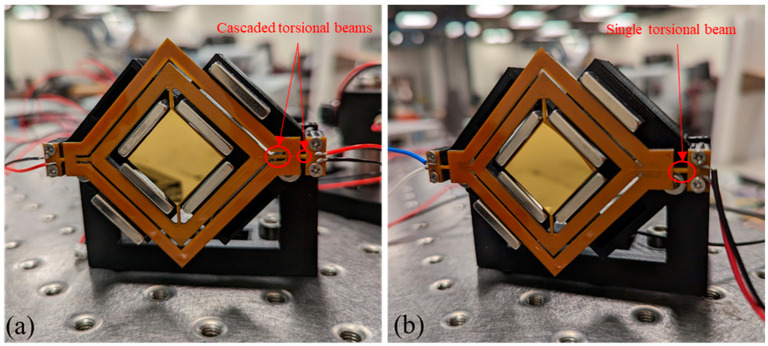
(**a**) Cascaded design prototype. (**b**) Traditional design prototype.

**Figure 8 micromachines-14-01954-f008:**
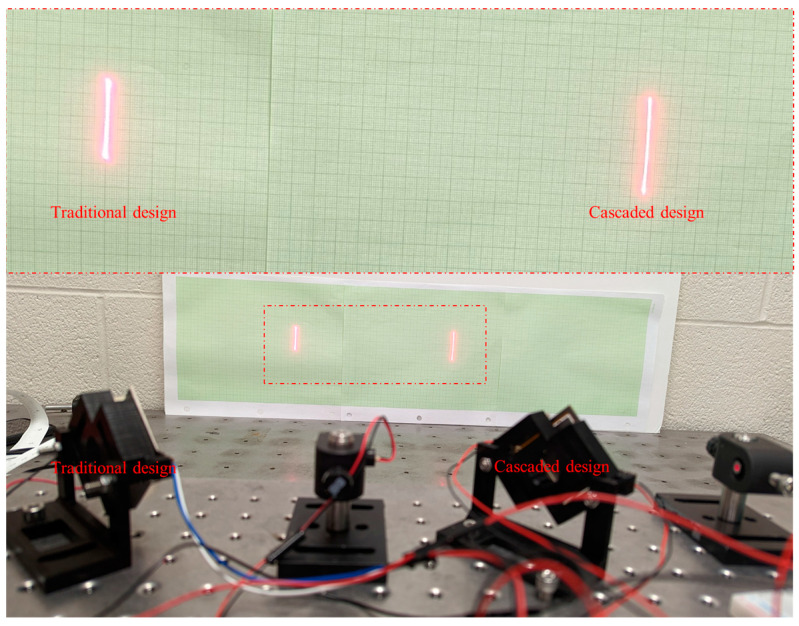
Vertical optical angle comparison between the traditional and cascaded design.

**Figure 9 micromachines-14-01954-f009:**
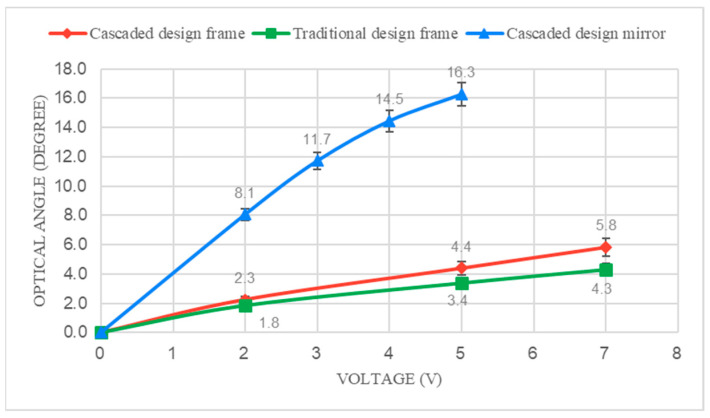
Optical angle vs. voltage diagram for the cascaded and traditional design.

**Figure 10 micromachines-14-01954-f010:**
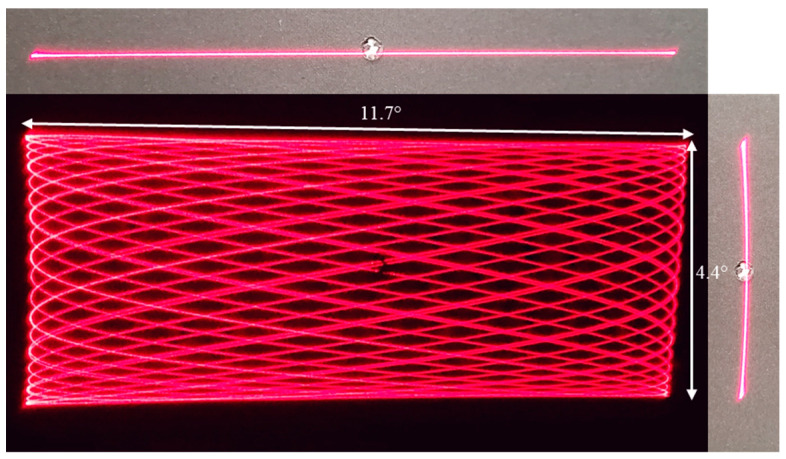
The Lissajous pattern generated by the cascaded design micromirror actuated at 290 Hz, 3 V horizontal scanning and 134.4 Hz, 5 V vertical scanning.

**Figure 11 micromachines-14-01954-f011:**
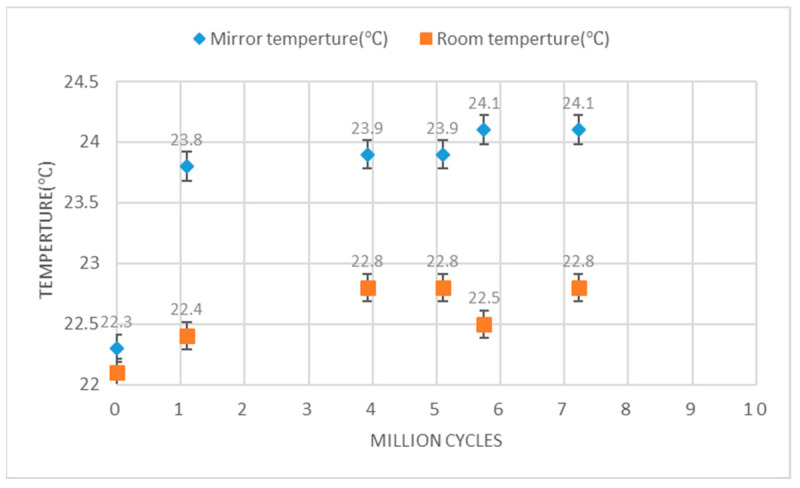
Temperature vs. actuation cycles during operation at 290 Hz with 5 V.

**Figure 12 micromachines-14-01954-f012:**
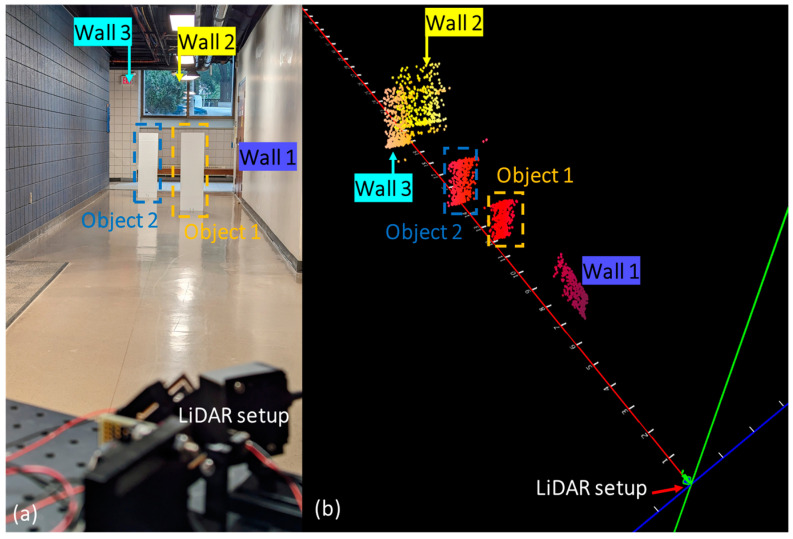
(**a**) LiDAR setup and scanned environment; (**b**) generated 3D point cloud.

**Table 1 micromachines-14-01954-t001:** Resonant frequency and mechanical static half-angle of the frame for the cascaded and traditional designs.

Frame Actuation Method	Resonant Frequency (f)	Mechanical Static Half-Angle of the Frame (θ)
Cascaded Design [22]	$f_{1}=\frac{1}{2\pi }\left(\frac{1}{2}\left(\frac{k_{1}I_{2}+{xk}_{1}\left(I_{1}+I_{2}\right)}{I_{1}I_{2}}\right)\right.$ ${\left.-\frac{1}{2}\sqrt{\left.{\left(\frac{k_{1}I_{2}+{xk}_{1}\left(I_{1}+I_{2}\right)}{I_{1}I_{2}}\right)}^{2}-4(\frac{{xk}_{1}^{2}}{I_{1}I_{2}})\right)}\right)}^{\frac{1}{2}}$	$\theta_{2}=\frac{T_{2}}{{xk}_{1}}+\frac{T_{1}{+T}_{2}}{k_{1}}$
Traditional Design	$f_{3}=\frac{1}{2\pi }\sqrt{\frac{k_{3}}{I_{3}}}$	$\theta_{3}=\frac{T_{1}+T_{2}}{k_{3}}$
Variables	Torsional stiffnesses: $k_{1},k_{2},k_{3}$ Stiffness ratio: $x=\frac{k_{2}}{k_{1}}$	
Constants	Torques: $T_{1},T_{2},T_{3}$ Mass Moment of Inertia: $I_{1},I_{2},I_{3}=I_{1}+I_{2}$	

**Table 2 micromachines-14-01954-t002:** The mass moment of inertia and torque values for the frame gimbals.

Symbol	Parameter	Value [Unit]
$I_{1}$	Mass moment of inertia of the outer frame	7.65 × 10^−7^ [kgm^2^]
$I_{2}$	Mass moment of inertia of the inner frame	4.53 × 10^−7^ [kgm^2^]
$T_{1}$	Torque generated on the outer frame	5.03 × 10^−6^ [Nm]
$T_{2}$	Torque generated on the inner frame	3.40 × 10^−6^ [Nm]

**Table 3 micromachines-14-01954-t003:** A comparison of characteristics of large aperture micromirrors.

Mirror	Mirror Plate Dimension or Diameter (mm)	Horizontal Optical Angle (rad)	Vertical Optical Angle (rad)	Horizontal Resonance Frequency (kHz)	Vertical Resonance Frequency (kHz)	FoM
Cascaded Design	19 × 19	0.28	0.08	0.29	0.13	0.55
Traditional Design	19 × 19	0.28	0.06	0.29	0.1	0.42
[16]	19 × 19	0.82	0.35	0.04	0.03	0.34
[30]	8	1.57	0.28	0.06	0.06	0.32
[31]	5 × 7.1	0.37	0.03	0.61	0.26	0.25
[32]	6.5	0.27	0.16	0.67	1.87	1.49
[21]	6	0.45	0.62	0.30	1.06	1.88
[33]	4 × 4	0.56	0.56	0.16	0.17	0.37

## Data Availability

The Zemax modelling data is available at: https://drive.google.com/drive/folders/15oANuBBxE6paoEovnY4USANwZQ_1hq-q?usp=share_link.

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
