# Peer review of "Cascaded 2D Micromirror with Application to LiDAR"

_micromachines, 2023, doi:10.3390/mi14101954_

Round 1

Reviewer 1 Report

This paper presented a large aperture 2D electromagnetic micromirror made of FPCB. The vertical scanning angle was increased by using a cascaded torsional beam design, and the micromirror was integrated into a 3D LiDAR system. This paper is interesting for large angle and low cost micro mirror. However, the authors are requested to make several modifications prior to the publications according to the following.

1. Section 2.1, please describe how the cascaded torsional beam micromirror works and explain why vertical angle amplification is achieved in detail. The variable f1、 f3、 θ2 、θ3 were not explained in the article. 

2. Please note the formatting and alignment of the formula, Eqution (1) in page 3 and Equation (2) in page 5.

 3. Generally, The optical aperture and optical element are round, why is the micromirror designed to be square?

 4. Is there any specific simulation or calculation of amplification ratio of cascaded torsional beam design? How about the quality factor ? Have these values been compared with the experimental results?

 5. In page 3 ,“In a traditional non-cascaded design, enhancing the generated torque necessitates increasing the coil's number of turns within the micromirror. However, this augmentation also increases the mass moment of inertia, causing a decrease in the resonance frequency…” Can you explain why does increasing the number of coils will reduce the resonant frequency? Increasing the coil's number of turns should not have a significant impact on mass quality, unless the volume is changed.

 6. As we know, laser has high energy, will the high energy cause damage on the mirror surface? Has the reflectivity of the micromirror changed after long-term operation? Otherwise, will the change in temperature affect the change in deflection angle, and how to control it?

 7. The optical angle was about 16° at 5V actuation voltage in Figure 9. However, the Lissajous pattern showed the angle was 11.7°, as shown in Figure 10. Please explain it.

 8. The figures are not clear enough, for example Figure 11 and Figure 12. Please use drawing software to draw the figures, and the coordinate format and font must be unified in the whole manuscript, such as Figure 3, Figure 9, Figure 13 and Figure 15.

Some mistakes in grammar and writing should be avoided, such as

“CNC “in the abstract should give the full name ,

“Micromirror/micromirror” should be consistent in the manuscript.

“The challenge associated with that mirror is that in order to increase the resonant frequency, the torsional stiffness should be increased”,”In this paper, the concept of cascaded torsional beams is adapted to the FPCB micromirror frame to help increase the scanning angle or increase the primary resonant frequency (and then the robustness) without lowering the scanning angle.” in page 2.……

Not limited to these, please check it carefully.

Author Response

The authors' reply to reviewer's comments can be found in the submitted file of \ "2653867 Response Letter .pdf"

Author Response

(The authors gave the same response as above.)

Round 2

Reviewer 2 Report

The authors have addressed all the points made in my first review report.

I still see typos. For instance,

page 11 (last paragraph): Horizontal => horizontal

Hope the errors will be corrected during the final production.